# Improving the Efficacy of Common Cancer Treatments via Targeted Therapeutics towards the Tumour and Its Microenvironment

**DOI:** 10.3390/pharmaceutics16020175

**Published:** 2024-01-26

**Authors:** Daniel Cecchi, Nolan Jackson, Wayne Beckham, Devika B. Chithrani

**Affiliations:** 1Department of Physics and Astronomy, University of Victoria, Victoria, BC V8P 5C2, Canada; danieldcecchi@uvic.ca (D.C.);; 2British Columbia Cancer-Victoria, Victoria, BC V8R 6V5, Canada; 3Centre for Advanced Materials and Related Technologies, Department of Chemistry, University of Victoria, Victoria, BC V8P 5C2, Canada; 4Division of Medical Sciences, University of Victoria, Victoria, BC V8P 5C2, Canada; 5Department of Computer Science, Mathematics, Physics and Statistics, Okanagan Campus, University of British Columbia, Kelowna, BC V1V 1V7, Canada

**Keywords:** cancer, chemotherapeutics, nanoparticles, radiotherapeutics, radiotherapy, chemotherapy, targeted therapies

## Abstract

Cancer is defined as the uncontrolled proliferation of heterogeneous cell cultures in the body that develop abnormalities and mutations, leading to their resistance to many forms of treatment. Left untreated, these abnormal cell growths can lead to detrimental and even fatal complications for patients. Radiation therapy is involved in around 50% of cancer treatment workflows; however, it presents significant recurrence rates and normal tissue toxicity, given the inevitable deposition of the dose to the surrounding healthy tissue. Chemotherapy is another treatment modality with excessive normal tissue toxicity that significantly affects patients’ quality of life. To improve the therapeutic efficacy of radiotherapy and chemotherapy, multiple conjunctive modalities have been proposed, which include the targeting of components of the tumour microenvironment inhibiting tumour spread and anti-therapeutic pathways, increasing the oxygen content within the tumour to revert the hypoxic nature of the malignancy, improving the local dose deposition with metal nanoparticles, and the restriction of the cell cycle within radiosensitive phases. The tumour microenvironment is largely responsible for inhibiting nanoparticle capture within the tumour itself and improving resistance to various forms of cancer therapy. In this review, we discuss the current literature surrounding the administration of molecular and nanoparticle therapeutics, their pharmacokinetics, and contrasting mechanisms of action. The review aims to demonstrate the advancements in the field of conjugated nanomaterials and radiotherapeutics targeting, inhibiting, or bypassing the tumour microenvironment to promote further research that can improve treatment outcomes and toxicity rates.

## 1. Introduction

The pandemic of increasingly common cancer-associated diseases has led to significant therapy advancements to treat these malignancies [1]. Surgery, chemotherapy, and radiation therapy are common treatments that may be prescribed to combat tumour growth as neoadjuvant/adjuvant or exclusive treatments [2]. Each patient’s treatment plan varies depending on numerous factors, such as the type of cancer and its local advancement [3]. While surgery is an effective method to remove large, localized tumour growths, chemotherapy and radiotherapy (RT) are often prescribed as adjuvant therapy to target microscopic cellular growth and metastases that cannot be removed via surgery. RT is one of the most common cancer treatment methods, where nearly 50% of patients who receive cancer treatment have RT in their regimen. A significant limitation of the current RT that limits effective dose escalation to the tumour is the unavoidable irradiation of surrounding healthy tissue. Advancements such as multi-leaf collimators and volumetric arc therapy [4] have improved the dose conformality and normal tissue toxicity; however, they cannot fully remove the potential for excessive damage to healthy tissue. 

Chemotherapy, a treatment methodology using cytotoxic compounds administered intravenously or orally, also has associated normal tissue toxicity, which significantly limits the therapeutic efficacy of the treatment and the total administered dose. For instance, Cisplatin may be prescribed for bladder or ovarian cancer but is commonly associated with severe kidney problems and gastrointestinal disorders [5]. Different classes of chemotherapeutic drugs include taxanes [6], antimetabolites [7], alkylating agents [8], topoisomerase inhibitors [9], vinca alkaloids [10], antineoplastic enzymes [11], and platinum agents [12], which can target various pathways in the tumour and its microenvironment [13]. As an example, topoisomerase enzyme inhibitors primarily target cells within the replication-S phase by inhibiting the topoisomerase enzyme and unwinding DNA strands, thus leading to DNA damage accumulation and cell death [9]. Meanwhile, taxanes, such as paclitaxel, docetaxel, and cabazitaxel, can cause G2/M cell arrest by stabilizing microtubules [6,14,15,16]. Of the total drug administered to the patient, it has been shown that only about 0.7% of the drug is delivered to the tumour, indicating the need for significantly greater systemic drug administration to deliver an adequate dose to the tumour [17]. Therefore, there is growing interest and research in combining common chemotherapeutic drugs with other nanomaterials and biomolecules to improve their cancer-cell-specific uptake and retention, thereby reducing normal tissue toxicity. 

Alternative to chemotherapeutic agents, radiosensitizers improving the therapeutic efficacy of radiotherapy cancer treatments have gained interest in past decades [18,19,20,21]. Radiosensitizers are compounds or molecules that, when combined with radiotherapy, improve the therapeutic effect of radiation by improving local dose enhancement or restricting malignant cells to radiosensitive phases of the cell cycle. Examples of radiosensitizers include high-Z NPs increasing local dose enhancement, oxygen promotors to improve the hyperoxia of the malignancy, and cell cycle restrictors prohibiting the progression of the cell cycle to remain in the radiosensitive phases. High-Z NPs have a high photoelectric cross-section (∝ Z^3^), making them particularly interesting for radiotherapeutic agents. Increasing the photoelectric cross-section, a greater number of low-energy electrons can be generated, termed “Auger” electrons [22]. These particles have a short range in tissue, leading to greater local dose deposition via their interaction with water, generating free radicals. In particular, the hydroxyl radical (OH-) is estimated to be responsible for two thirds of the X-ray damage to DNA in mammalian cells via abstraction of the deoxyribose hydrogen atom [23]. Other radiosensitizers can work by blocking the progression of the cell cycle, inhibiting critical DNA repair. Additionally, they may restrict progression to the G2/M phase—the most radiosensitive phase of the cell cycle. The combination of NPs along with cell-cycle-targeting therapeutics may offer a unique advantage in combating normal tissue toxicity by significantly improving therapeutic efficacy. A critical and necessary step in applying radiosensitizers is their specific uptake in the target volume. Their functionalization with other nanomaterials and targeting ligands could improve cell-specific uptake and bioavailability within the tumour. If not appropriately functionalized, the opsonin protein within the blood will absorb the NPs and be removed from the circulatory system by macrophages [24,25]. Polyethylene glycol (PEG) has been demonstrated in numerous studies to be an effective conjugation to NPs to increase the blood circulation time, which subsequently contributes to a greater possibility of tumour penetration via the enhanced permeability and retention (EPR) effect [26,27,28,29,30,31]. EPR is the process by which small molecules preferentially invade tumour tissue based on its poor structural integrity and leaky endothelial conjunction [32,33]. 

One of the most significant contributors to the tumour’s immunosuppressive ability, proliferation, and metastatic potential is the tumour microenvironment (TME) [34]. Section 2 discusses various cellular and non-cellular components of the TME, including pro- and anti-immunogenic factors, excreted growth factors, proteins, and other pro-tumourigenic cells [35,36,37,38,39,40]. Each of these components and their respective pathways offers a unique opportunity for specific targeting to inhibit tumour growth and improve drug and nanoparticle uptake within the tumour. Current chemotherapy agents already use some of these pathways to elicit cellular damage and tumour-specific drug uptake [13]. However, there is a growing demand to improve the targeting of these pathways, not only for improved local control but also to reduce normal tissue toxicity rates. Targeting components within the TME may achieve the improved cellular uptake of nanoparticle radiosensitizers and other cytotoxic drug compounds (Figure 1). 

Radiosensitizers and other therapeutic agents aim to improve the therapeutic index (TI) by reducing the required dose of a particular agent to elicit damage to the tumour (Figure 1). As such, extensive time and effort have been devoted to the research of anti-cancer drugs and radiosensitizing agents to improve treatment outcomes and reduce normal tissue toxicity. The intricacy of nanomaterial development requires significant research to optimize their size, functionalization, target, etc. For example, the cytotoxicity of nanomaterials greatly inhibits their application in clinical studies [41,42,43,44]. Additionally, the aggregation of NPs can be beneficial in increasing the particle size [45], but may also decrease permeability within tumours. Other NPs may exhibit magnetic or electrical properties affecting their biodistribution and biocompatibility [46]. For instance, gadolinium-based NPs are routinely used to enhance magnetic resonance imaging but can have associated toxicity in the event of release [47]. Other issues arise, such as aggregation due to surrounding potentials and difficulty controlling their performance, which may be affected by their magneto-electric properties. Given the complexity of NP formulation, extensive research is put forth prior to in vivo tests and clinical trials. 

With the complexity of the tumour and its TME, its ability to obstruct treatment pathways, and the propensity for the tumour to adapt to treatments, there is a requirement for further research and effort to improve drug delivery and toxicity rates. This review article discusses the current literature regarding current and developing chemotherapeutics and nanotechnology to target or circumvent the TME. Given the complexity of the TME exploited by the tumour to proliferate and metastasize, there are multiple promising avenues to develop targeted therapy. The goal of this article is to demonstrate the significant advancements of conjugated nanomaterials and radiosensitizers as effective anti-cancer drugs by their active targeting and evasion of the TME. While the lists of conjugated particles in the following sections and the corresponding ligands currently in review and clinical application are not exhaustive, they serve as a demonstration of the improvement gained in the therapeutic efficacy of cytotoxic compounds via specific targeting molecules. 

## 2. Pivotal Contributors of the Tumour Microenvironment for Tumourigenesis

Carcinogenesis is a complex process that involves the accumulation of genomic and molecular mutations in healthy cells. These mutations can promote tumour growth and donate tumour resistance to anti-cancer drugs and the body’s innate immune response [48,49]. Characterization of these mutations can be achieved through many tools, such as fluorescent in situ hybridization (FISH) [50], optical genome mapping (OGM), or specific cancer detection strategies such as Xpert^®^ Bladder-Cancer Detection [51]. FISH is an encompassing technique describing the use of fluorescently labeled DNA probes, enabling researchers to observe the gene distribution. Its application in cancer diagnosis is well documented and demonstrates its reliability in cancer research [52]. OGM, a technique similar to FISH, provides a broad overview of the genome, detecting structural genetic variations. Similarly, OGM also stands out as an effective sequencing technique for cancer compared to other methods such as FISH due to its ability to detect karyotypes—a complete set of chromosomes [53,54]. 

Surrounding and penetrating the tumour structure, the tumour microenvironment (TME) is largely responsible for shaping the aggressiveness of the cancer, influencing the treatment response and metastatic potential (Figure 2) [55,56,57]. By exploiting the complex network of the TME, the tumour elicits bidirectional communication between its malignant and non-malignant cells to infiltrate the surrounding tissue, proliferate, and metastasize. Preferentially targeting these pathways with patient-specific and tumour-specific treatments can limit tumour growth and spread and elicit apoptosis [58]. For instance, the targeting of cancer-associated fibroblasts (CAFs) has been achieved with the anti-cancer compound Minnelide—a diterpene triepoxide—or with Pirfenidone—a synthetic pyridone drug [59,60]. The targeting of endothelial cells was also achieved by suppressing VEGF-VEGFR2 in ovarian cancer cells, which inhibited angiogenesis [61]. Many other reviews have been published discussing the TME and its various components [34,35,36,38,57,62,63,64,65,66]; the following discussion will serve to demonstrate the complexity of the TME and important contributors to the tumour’s immunosuppression and evasion. 

During tumour progression, the TME evolves to circumvent normal cellular interactions that would otherwise be detrimental to tumour growth [67]. For instance, during early tumour development, the tumour primarily relies upon diffusion to gain oxygen and other nutrients. However, when the tumour reaches approximately 1–2 mm^3^, diffusion can no longer sustain the nutrient requirements; therefore, the TME becomes hypoxic and acidic from metabolic waste and an insufficient oxygen supply. Through transcribed hypoxia-induced factor-1α (HIF-1α) [68], the TME instructs endothelial cells to produce proangiogenic factors such as platelet-derived growth factor type B (PDGF-B), vascular endothelial growth factor (VEGF), and hepatocyte growth factor (HGF), among others, to stimulate the formation of new blood vessels to facilitate growth and ensure that an adequate supply of nutrients is provided [69]. 

A critical step in tumour progression is the evasion, suppression, and transformation of the immune system to work towards tumourigenesis [70]. T and B cells, natural killer (NK) cells, macrophages, neutrophils, and dendritic cells occupy the body’s innate and adaptive immune response [67]. As part of the immune system, macrophages can have both an innate tumour-suppressing (M1 stage) and tumour-promoting (M2 stage) phenotype [71]. Tumour-associated macrophages (TAMs), the M2 phenotype, are promoted within the tumour through hypoxia and the secretion of factors such as IL-4 [64]. Tumour-associated neutrophils (TANs) can also become recruited by the TME to contribute to angiogenesis by the secretion of VEGF and can induce immunosuppression in T cells by PD-L1 expression [72,73,74]. Myeloid-derived suppressor cells (MDSCs) also develop within the TME to promote immune evasion and tumour vascularization, where they have been shown to inhibit T cell activation, NK cell cytotoxicity, and antigen presentation by dendritic cells [75]. 

The hypoxic state of the tumour is critical for its anti-radiotherapy ability and proliferation and metastatic potential [76,77]. The concentration of reactive oxygen species (ROS) is generally regarded as detrimental to the tumour due to their highly reactive nature. As Saikolappan et al. [78] discussed, ROS also play a pivotal role in tumour growth and metastasis, leading to an inherent balance between excessive ROS that damage tumour cell DNA and sufficient ROS to promote tumour proliferation [79]. The present hypoxia-induced factor-1 (HIF-1) is largely responsible for mediating the effect of hypoxia on tumour cells. Research has shown that it encodes multiple growth factors, such as VEGF, PDGF-B, TGF-β, and transferrin receptors and anti-apoptotic factors, among others found in the TME [68,80]. The hypoxic cycle is characteristic of most tumours. It is defined by the cycling of hypoxia and reoxygenation of localized tissue due to the heterogeneity of the tumour and abnormal vascular networks [81]. The plethora of excreted factors, proteins, etc., by HIF-1α subsequently induces angiogenesis, but the leaky vasculature of tumour endothelial cells (TECs) causes a corresponding decrease in angiogenesis, and the cycle arises. By targeting HIF-1α and other factors that either promote tumour hypoxia or contribute to the hypoxia cycle, the tumour itself may be halted in a state of oxygenation, thereby increasing the number of ROS during radiation and subsequent DNA damage.

The TME transforms normal cellular processes throughout tumour progression to promote angiogenesis and deliver nutrients and oxygen to the tumour. Effective targeting of the TME can promote the body’s natural immune response and inhibit regulatory pathways used by the tumour, effectively starving the malignancy and obstructing its metastatic potential. Passive targeting with NPs and drug conjugates takes advantage of the EPR effect to uptake within the tumour tissue as opposed to healthy tissue [18,82,83]. In contrast, active targeting of the TME involves specifically functionalizing the nanomaterials to bind to receptors overexpressed by TME components. In the following sections, advancements in both the active and passive targeting of the TME will be discussed. 

## 3. Targeting the Tumour Microenvironment with Conjugated Chemotherapeutics and Nanomaterials

This section highlights how the targeting of the multiple TME pathways is achieved with conjugated nanomaterials (Figure 1). Using clinically relevant chemotherapeutic agents conjugated to ligands and other compounds, the drugs can be selectively deposited near or within the tumour, reducing their cytotoxic nature to the surrounding healthy tissue. Alternative to the cytotoxic chemotherapeutic compounds, other molecules designed to specifically inhibit communication pathways within the TME will be discussed. The following sub-sections will highlight the targeting of key cellular (ECs, CAFs, TAMs) and non-cellular (ECM) TME components based on the current developments of NP-based therapeutic options (Table 1). 

### 3.1. Endothelial Cells

Endothelial cells line the inside layer of blood vessels and regulate exchanges between the blood vessels and the surrounding tissue. TECs are functionally similar, though they participate in the active promotion of tumour growth and metastasis through tumour angiogenesis. Therefore, anti-angiogenesis treatments through the targeting of TECs have the potential to significantly limit tumour growth by inhibiting the tumour’s blood supply. Certain proteins’ expression within TECs allows for targeted treatment, distinguishing them from normal endothelial cells. For instance, the gene lysol oxidase (LOX) was found by Osawa et al. [101] to be upregulated within TECs and contributed to tumour metastasis and invasion. Other markers like suprabasin [102] and cyclooxygenase-2 (COX-2) [88] have also been shown to be potential targets within TECs for anti-angiogenesis treatments. 

For specific TEC targeting, a common peptide used is arginine–glycine–aspartic acid (RGD), which preferentially binds to the α(v)β(3) integrin—highly expressed in TECs [103]. A study by Danhier et al. [15] used PEGylated paclitaxel (PLX) grafted with RGD to preferentially target the tumour endothelium based on the expression of the α(v)β(3) integrin. PLX (commonly known as Taxol©) is an anti-neoplastic agent that is commonly used to treat ovarian or breast cancers. Its conjugation with the RGD ligand showed improved cellular uptake and retention within tumour models and corresponded with an improved survival rate of tumour-bearing mice when compared to non-grafted paclitaxel. The PEGylation of therapeutics has been demonstrated to significantly improve the blood circulation time of NPs and is commonly employed in the discussed literature in this review [104]. Other ligands that have been shown to preferentially bind to markers within TECs are the peptides asparagine–glycine–arginine (NGR) and Cys–Gly–Lys–Arg–Lys (CGKRK). NGRs have shown high affinity towards CD13 (aminopeptidase N), which is also overexpressed within TECs. Their targeting ability has been demonstrated to improve the cellular uptake and cytotoxicity of the therapeutic compound doxorubicin (DOX) (Figure 3a) [85]. The dual conjugation of NGRs with a cell-penetrating peptide (CPP, CGRRMKWKK) has also been shown to have greater intracellular uptake, demonstrating the cooperative effects of dual-targeting ligands [86]. CGKRK peptides preferentially target heparin sulfate, located on the surfaces of TECs. Their conjugation with PLX, and their combination with Pep-1—a targeting ligand for the IL-13Rα2 receptor on glioma cells— demonstrated significantly improved anti-glioma efficacy with negligible acute toxicity [87]. To improve the intratumoural uptake of lipid NPs, Sakurai et al. [84] used a cyclic-RGD-modified liposomal siRNA (RGD-MEND) on human renal cell carcinoma (RCC), OS-RC-2-bearing mice, which preferentially targeted TECs and not normal ECs [105], to silence VEGF-2. The inhibition of VEGF-2 correlated with increased lipid NP accumulation within the tumour (Figure 3b). 

As mentioned previously, the EPR effect is largely responsible for bionanotechnology distribution to tumours. However, given that this effect is not present with smaller, less mature tumours, techniques to elicit endothelial leakiness (EL) may prove beneficial for earlier cancer treatments. A study by Wang et al. [106] attempted to induce EL by reason of negatively charged GNPs that target EC membranes. Their results demonstrated that negatively charged GNPs introduced greater gaps in the cell membrane compared to positively charged GNPs (Figure 3c). The authors concluded that this effect could be beneficial in eliciting EL in tumours that do not have an intrinsic EPR pathway. Conversely, enhancing the EPR effect with NPs can have negative consequences such as the promotion of tumour metastasis [107]. To address this concern with the application of NPs, Huang et al. [89] developed PLGA-ICG-PEI-Ang1@M NPs—a PLGA core and inner shell of positively charged polyethyleneimine and anti-permeability growth factor Angiopoietin 1 (Ang1), coated with a Jurkat cell membrane. Upon drug uptake within the endothelial layer, the cell membrane on the NP surface is ruptured, releasing Ang1, which subsequently restores the disrupted endothelial layer. In vitro results demonstrated reduced cancer cell migration while also recovering endothelial cell leakiness.

### 3.2. Cancer-Associated Fibroblasts

The separate phenotype of CAFs from normal fibroblast cells is characterized by their ability to interact with CAFs and participate in tumour growth and metastatic potential through the release of growth factors such as tumour growth factor β (TGF-β), hepatocyte growth factor (HGF), and CXC chemokine ligand-1, which binds to the CXCR4 ligand expressed on the surfaces of tumour cells [40,108]. CAFs are a critical component of the stromal tissue of tumours and aid in nanoparticle capture, leading to a decrease in the overall effectiveness of chemotherapeutics. Therefore, significant research has been directed to developing nanoparticle-conjugated therapeutics to bypass the CAF checkpoint, thereby improving drug delivery to the tumour itself [109]. 

Sitia et al. [110] targeted CAFs using H-ferratin nanocages functionalized with fibroblast activation protein (FAP) antibody fragments. Navitoclax, an experimental pro-apoptotic drug, was encapsulated within the NPs. The NP demonstrated significantly higher binding to FAP-positive CAFs than FAP-negative cancer cells, which corresponded with greater drug release and toxicity from navitoclax. A polymer conjugate of docetaxel (DTX), PEG, and acetylated carboxymethylcellulose (Cellax-DTX) was evaluated for its effect on smooth-muscle-actin-positive CAFs in pancreatic cancer xenografts and a metastatic PAN02 pancreatic cancer mouse model [93]. The authors reported that greater than 90% of Cellax-DTX accumulated within the CAFs and resulted in the long-term depletion of the stromal cell population (Figure 4a,b). By depleting the population of stromal cells, they found a greater than 10-fold increase in tumour perfusion, reduced tumour weight, and decreased metastasis. While moderate toxicity was reported in the mice via observed weight loss, the effects disappeared after 10 days. Other receptor proteins that have been studied for targeting by NPs include the FAP receptor via a peptide NP loaded with DOX and with a mouse monoclonal antibody (mAB) modified to the surface (PNP-D-mAB) by Ji et al. [90]. The NP-conjugated DOX demonstrated a significantly improved tumour penetration capability by the depletion of CAFs and breakage of the stromal barrier. Tumour growth was subsequently greatly reduced compared to non-coated DOX, PNP alone, and antibody immunoglobulin G NPs (PNP-D-IgG). The lipid anisamide has also been demonstrated to improve the uptake efficiency by targeting sigma receptors on the surfaces of CAFs. Miao et al. [91] developed lipid-coated calcium phosphate NPs with the lipid anisamide conjugated to the surface. With the addition of the anisamide, uptake within CAFs was ~seven times greater than in other cells (Figure 4c). CAFs also express PDGF receptors. Bansal et al. [92] joined anti-cancer agent IFNγ to a PDGF-β receptor conjugated with human serum albumin (PPB-HSA-IFN-γ). Their results demonstrated the complete abolishment of advanced liver cirrhosis in mice, with the minimal accumulation of the NP in other organs such as the kidney, heart, and lungs 15 min after intravenous injections. 

### 3.3. Immune Cells

Contained within the TME are immune suppressor and pro-immunogenic cells such as neutrophils, CD8+ cytotoxic T cells, regulatory T cells (Tregs), and MDSCs (including macrophages and dendritic cells), which are targets for immunotherapy. CD8+, M1 macrophages, and N1 neutrophils participate in the anti-tumour response through the secretion of factors such as INF-γ and other cytokines and the suppression of NK cells and T cells [39]. Neutrophils are bisected into two phenotypes: pro-tumourigenic N2 cells and tumour-inhibiting N1 cells. N1 neutrophils secrete factors such as tumour-necrosis factor-α (TNF-α), nitric oxide, and antibody-dependent cellular cytotoxicity [111,112]. Factors such as TGF-β (secreted by CAFs) help to polarize N1 neutrophils to N2 neutrophils, with the inverse being true as well [113]. Macrophages are, too, bisected into two phenotypes, M1 and M2. M1 phenotypes participate in anti-tumour immune processes, while M2 phenotypes help to promote tumour growth and metastasis [71,114]. During the early stages of tumour growth, the tumour secretes chemokines (CCL2, for example) and other growth factors (epidermal growth factor, for example [71]) that attract and convert M1 macrophages to M2. Once converted, factors such as IL-1Ra, an immunosuppressive cytokine, and CCL5, a pro-metastatic chemokine, are upregulated by TAMs. These components offer unique pathways for targeting with NPs and other compounds to limit the metastatic cascade and tumour proliferation [115]. CD8+ cells constitute the primary anti-tumour immune response; however, they can become exhausted in the TME, leading to their disfunction and unsuccessful clearing of pathogens such as malignant cells. Their upregulation by immune-supportive biomolecules, therefore, makes them a promising candidate for anti-tumour targets. 

NPs that can preferentially bind to and target various pro- and anti-immunogenic TME components have been developed. Targeting the upregulation of CD8+ T cells has proven effective in increasing the immune response to malignant growths. Oberli et al. [97] developed lipid NPs (B-11) to deliver mRNA vaccines that encoded the model immunology protein ovalbumin (OVA) to B16-OVA melanoma-containing mice. Upon administration, CD8 T cell proliferation was increased and these proliferated CD8 T cells correlated with reduced tumour growth compared to untreated mice (Figure 5a,b). Targeting chemokines that recruit M1 macrophages with NPs is one avenue currently being explored to limit the aggregation of M2 macrophages. NPs such as 7C1 loaded with chemokine-CX3Cl1 targeting ligands (7C1-Axo-siCX3CL1) successfully reduced the recruitment of macrophages within the tumour region, as well as inhibiting tumour growth [94]. One study by Zang et al. [95] developed lipid-coated calcium zoledronate nanoparticles (CaZol@ pMNPs) to target the surface marker CD206 on TAMs. The authors reported reduced angiogenesis and inhibited the tumour’s natural immune suppression, which limited tumour growth with no significant changes in surrounding organs. To target M2 macrophages, Zeng et al. [116] developed imidazole- and mannose-modified carboxymethyl chitosan nanoparticles (MIC-NPs). Imidazole is an apoptosis-inducing drug with remarkable anti-tumour capabilities. Through its conjugation with mannose-modified carboxymethyl chitosan, M2 macrophage recruitment was significantly lower than that in the control group (*p* < 0.05). Another potential treatment avenue is the reprogramming of M2 macrophages back to the M1 phenotype. Shan et al. [98] developed a nanoparticle peptide carrier system, M2pep-rHF-CpG, targeting M2 macrophages via TLR9, the receptor for CpG oligodeoxynucleotides. The results demonstrated the effective repolarization of M1-like TAMs in vitro and in vivo and reduced tumour development in a 4 T1 tumour-bearing animal model. Additionally, their results demonstrated no significant change in mice body weight with the novel NP compared to a saline control, nor did it affect inflammatory cytokines within the blood. The targeting of neutrophils with NPs was achieved by Vols et al. [96]. The authors used PLGA nanoparticles and liposomes decorated with a CD177-targeting peptide LQIQSWSSSP tetramer (denoted LQI tetramer) and demonstrated the significant targeting of neutrophils without functional disruption (Figure 5c). Another study by Li et al. [117] developed cisplatin-loaded nano-pathogens to hitchhike neutrophils to target residual microtumours. Their research reported significant tumour shrinkage during photothermal therapy within mice, demonstrating the potential use of nano-pathogens for increased tumour targeting. 

### 3.4. Extracellular Matrix

The ECM consists of various non-cellular components that participate in tumour progression, including glycoproteins, collagen proteins, elastin, and elastic fibres, among other components. Throughout tumour progression, the ECM evolves to recruit healthy cells to benefit the tumour, which secrete additional proteins, enzymes, and cytokines [118]. Specifically, CAFs are one of the most important drivers of ECM growth by the secretion of collagen proteins—the largest contributor to the ECM composition [119]. Collagen proteins are classified into four families according to their supramolecular assemblies [120]. Collagen can connect to cancer cell receptors like DDR1 and DDR2 [121,122] as well as to cellular adhesion molecules like integrin, which can affect various signalling pathways, like the MEK/ERK pathway, inducing the proliferation and invasion of squamous cell carcinoma [123]. Other ECM components include fibronectin, hyaluronic acid, and laminin. Fibronectin, though not as abundant within the ECM as collagen, still plays an important role in tumour progression through its interaction with various growth factors, like VEGF [124], TGF-β [125], and PDGF [126], and its interaction with integrin α(v)β(6) [127]. Due to the production of collagen and other proteins, growth factors, cytokines, etc., the ECM surrounding tumours becomes stiff, which can also limit the penetration effect of anti-tumour drugs. 

Targeting of the stromal thickness has been achieved with Tranilast (N-(3,4-Dimethoxycinnamoyl)anthranilic acid), an anti-fibrotic drug. As discussed by Osman et al. [128], the mechanisms of action of Tranilast can vary widely, from inhibiting angiogenesis to the suppression of TGF-β and other key intracellular pathways. Saini et al. [129] used the molecule in combination with doxorubicin—a clinically relevant chemotherapeutic drug—to reduce the stiffness of the stromal matrix within 3D tumour organoid models. The combinatory effects of the two compounds disrupted fibronectin assembly and the collagen fibre density. 

Collagen constitutes a significant portion of the ECM, which makes it an ideal target to reduce the stiffness of the ECM, thereby enabling the greater penetrating power of chemotherapeutic agents. Collagenase, an enzyme that can break down collagen, may enable drug–tumour penetration. Collagozome is a proteolytic enzyme NP that encapsulates collagenase type-1 by 1,2-dimyristoyl-sn-glycero-3-phosphocholine. Zinger et al. [99] demonstrated that pancreatic ductal adenocarcinoma (PDAC) pre-treated with collagozome and treated with PLX showed an 87% reduction in tumour size compared to PDAC pre-treated with liposomes. In addition, after treatment with collagozome, the injected GNPs demonstrated significantly improved biodistribution in healthy and malignant tissue (Figure 6a). However, as discussed by Huang et al. [35], there are concerns with regard to the application of collagenase in cancer therapy given the growth factors secreted during the degradation of collagen, which could promote tumour progression. 

Extra domain A (EDA) and extra domain B (EDB) of fibronectin are often upregulated in the tumour vasculature, which can lead to specific treatments enabling the targeting of these proteins for drug delivery [100,130]. Saw et al. [100] labelled DTX-containing micelles with a bipodal aptide targeting EDB (APT_EDB_-DSPE-DTX). Their results demonstrated improved anti-tumour efficacy towards malignant glioma cells with the targeting nanoparticles compared to non-targeted controls (Figure 6b). In addition, a 20% reduction in free DTX was found, correlating with the improved glioma targeting of APT_EDB_. APT_EDB_ has also been conjugated with PEG-PLA loaded PTX (APT-NP-PTX) by Gu et al. [14] to target malignant gliomas. The conjugation with APT_EDB_ showed the improved cytotoxicity of PTX on U87 MG cells, improved anti-glioma efficacy, and elevated cellular internalization compared to unmodified NPs. 

## 4. Radiopharmaceuticals as Local Dose Enhancement Agents

The following sections will discuss the effect of radiosensitizers conjugated with NPs and other targeting ligands on the tumour and its microenvironment. As depicted in Figure 1, chemotherapeutics, nanomaterials, and radiosensitizers can be functionalized in unison for improved tumour and TME targeting. In particular, the following non-exhaustive list of classes of radiosensitizers will be explored: biomolecules that improve the oxygenation of tumour growths; compounds that interact directly with radiation, such as GNPs; and compounds to affect cell cycle distribution, such as taxanes. Similar to Section 3 above, Table 2 lists several radiopharmaceuticals and their respective mechanisms of action to improve the local dose. 

### 4.1. DNA Damage Stabilizers and Hyperoxia-Inducing Radiosensitizers

The hypoxic state of the tumour itself leads to radioresistance. In normal tissue, the consumption and supply of oxygen are stable, while, in tumours, this stability is broken. A hypoxic environment leads to hypoxia-induced factors (HIF) such as HIF-1α and HIF-1β, which induce the production of VEGF and thus tumour progression and angiogenesis [143]. The hypoxic state leads to reduced ROS and less damage to DNA during radiation exposure. Interestingly, despite being under hypoxic conditions, the tumour generates ROS at a greater rate than normal tissue [144]. As discussed by Wang et al. [145], the tumour, therefore, relies heavily on an antioxidant system to keep the concentration of ROS at sublethal levels. Therefore, targeting these pathways within the tumour and TME that control the level of ROS can both restrict the blood supply to the tumour, limiting tumour growth, and increase oxygenation within the tumour for greater ROS generation and radiosensitization. 

One compound, nitric oxide (NO), has been shown to improve the tumour vasodilation and generation of ROS and nitrogen species (RONS), which could improve the radiosensitization of the malignancy [146]. Vasodilation has correlated tumour-promoting effects and increasing blood flow, which increases the tumour growth potential. In addition, the inhibition of NO synthesis within the tumour environment has been demonstrated to reduce the tumour blood volume, which can correspondingly reduce tumour growth [147]. As Scicinski et al. [148] have previously discussed that NO does show potential as an effective radiosensitizer along with oxygen to improve tumour oxygenation, its corresponding contradictory effects by promoting tumour vascularization necessitate further research on NO-mediated radiosensitization. To this end, Zhang et al. [131] developed a bismuth-based nanotheranostic agent, which was functionalized with S-nitrosothiol (Bi-SNPs), for use with near-infrared photothermal therapy and radiation therapy. In vivo experiments were performed on HepG2 cells incubated with or without Bi-SNPs and subsequently irradiated with X-rays to 5 Gy. Via a DNA DSB assay, the authors reported significantly increased DNA damage with Bi-SNPs+RT. In addition, the level of HIF-α expression was reduced following the administration of NPs and treatment. Lastly, minimal toxicity was observed in mice, indicating good biodistribution and potential for biomedical applications in future treatments. 

Improving tumour oxygenation by either direct exogenous oxygen delivery or oxygen generation in situ could also directly benefit radiosensitization and ROS generation [18]. For instance, Lu et al. [132] developed hollow mesoporous organosilica NPs encapsulating hydrophobic perfluoropentane and conjugated with CuS (HMCP). The surface of the NPs was then PEGylated (PFP@HMCP) for an improved circulation half-life. Oxygen-saturated PFP@HMCP (O_2_-PFP@HMCP), after near-infrared radiation (NIR), demonstrated a high-level oxygen concentration within U87MG cells. Concurrently with NIR and RT, O_2_- PFP@HMCP was able to eradicate all tumours within 20 days (Figure 7a). Song et al. [133] fabricated PEGylated perfluorocarbon nanodroplets decorated with TaOx nanoparticles (TaOx@PFC-PEG). In mice bearing 4T1 tumours, a significant improvement in tumour oxygenation from ~10% to ~37% was reported post-injection of TaOx@PFC-PEG, which was greater than that for TaOx-PEG alone. After 6 Gy irradiation, the tumour growth delay was significantly improved in mice treated with TaOx@PFC-PEG compared to TaOx-PEG+RT or RT alone. 

Proposed in 2006 by Professor Ogawa from Kochi University in Japan, Kochi Oxydol Radiation Therapy for Unresectable Carcinoma (KORTUC) is a novel radiosensitizer that has recently been investigated for its application with interstitial brachytherapy [134]. The function of KORTUC is to maintain the oxygen levels within malignant tissue containing larger amounts of hypoxic cells or anti-oxidative enzymes. Hydrogen peroxide (H_2_O_2_) is the active ingredient in KORTUC, with sodium hyaluronate acting as a stabilizer for H_2_O_2_, to maintain oxygen levels as well as to target the CD44 receptor ligand [149]. H_2_O_2_ has been demonstrated to not only produce oxygen within tumour tissues but also to act as an agent to deactivate antioxidant enzymes [149,150]. Kemmotsu et al. [151] applied KORTUC to mouse models that were injected with murine tumour cells at two distinct locations. KORTUC + RT significantly reduced tumour growth compared to RT alone or no RT (Figure 7b). Shimbo et al. [134] compared results from patients treated with KORTUC and interstitial brachytherapy since 2012 for locally recurrent cervical cancer and found a significantly improved 2-year local control rate between patients with prior external beam RT and those with no history of RT (*p* = 0.02 (Figure 7c). The patients were injected intratumourally with KORTUC, which can preferentially accumulate within the tumour rather than healthy tissue. No adverse side effects from intratumoural KORTUC injections were reported. 

### 4.2. High-Z Nanoparticles

Improving the interaction cross-section of ionizing radiation with malignant tissue is another avenue that is actively being explored to improve current RT treatments [18,83,152,153]. High-Z metal NPs increase the photoelectric cross-section of a tissue, within which they preferentially attenuate. Common inorganic radiosensitizers that have been employed are gold-based NPs [19], gadolinium-based NPs [135], silver-based NPs [137], lanthanide-based NPs [154], and platinum-based NPs [155]. Given their strong affinity towards local dose deposition, there is a requirement that these NPs be conjugated with specific targeting ligands associated with receptors characteristic of malignant cells to limit normal tissue dose enhancement. Typically, metal NPs are employed to target the malignant tissue itself; however, given the increasingly expanding role of the TME throughout tumour progression, conjugating high-Z NPs with TME-targeting ligands may also be an important pathway to limit tumour growth and metastasis. 

Given the complexity of the TME and the tumour itself, there exist multiple pathways by which NPs can preferentially target to improve endocytosis [156]. For instance, via the EPR effect, small 3-nm hydrodiameter gadolinium-based NPs, termed AGuIX, were endocytosed within carcinoma H1299 cells [135]. The study reported a significant tumour growth delay in mice implanted with the H1299 cells from irradiation between 0 and 6 Gy. AGuIX as NPs are not specifically conjugated with additional targeting ligands, and their cancer-cell-specific uptake is apparently attributed to the EPR effect [157]. A study by Ho et al. [158] took this further by conjugating polyacrylic-acid-coated Gd_2_O_3_ NPs with either cyclic RGD (cRGD) or folic acid (FA), or both. As described above, RGD specifically targets the α(v)β(3) integrin receptor within the TME, while FA targets overexpressed folate receptors within the tumour itself, facilitating NP penetration [136]. The presented results by Ho et al. demonstrated minimal cytotoxicity to normal cells up to a 500 μM Gd concentration, with increased toxicity observed with increasing Gd concentrations within U87MG cells. The multi-targeting capabilities of conjugating both cRGD and FA slightly improved the toxicity of the NPs against U87MG cells compared to FA alone (Figure 8a). The study did not evaluate the specific uptake of the Gd-NPs for their effect on radiosensitivity; however, it demonstrates the tumour-specific uptake of a high-Z metal NP using targeting ligands that could translate to an improved therapeutic index during radiotherapy. 

Using Ag-NPs conjugated with PEG (PNP) or aptamer As1411 (AsNP), Zhao et al. [137] demonstrated improved tumour-specific uptake and targeting. The incubated C6 glioma cells were irradiated between doses of 0 and 8 Gy from a 6MV LINAC. Using clonogenic assays, the authors reported a sensitization enhancement ratio for Ag-NPs, PNPs, and AsNPs of 1.22, 1.31, and 1.62, respectively (Figure 8b). In glioma-bearing mice, mean survival times also increased from 24, 30, 35, and 45 days when treated with saline, Ag-NP, PNPs, and AsNPs at a dose of 10 mg/kg and a dose of 6 Gy per mouse. In an attempt to replicate a more realistic tumour environment by incorporating the extracellular matrix, Bromma et al. [138] evaluated three-dimensional tumour spheroids for GNP+DTX uptake. The synthesized GNPs were PEGylated and conjugated with RGD. Their results demonstrated a significant GNP concentration gradient in the 3D spheroids, along with increased GNP uptake with the use of DTX. Recently, Alhussan et al. [139] have also demonstrated the significant intratumoural uptake of GNPs mediated by DTX via CAFs. The spherical GNPs were subsequently functionalized with PEG and RGD for improved tumour-targeting and retention capabilities. Interestingly, with the employed PEG and RGD functionalization, CAFs demonstrated a ~7-fold increase in NP uptake compared to tumour cells; however, tumour cells had improved NP retention. The improved NP retention was attributed to two factors: (1) DTX arrests the cell cycle, inhibiting the division of GNPs from one cell to two; (2) the stabilization of microtubules significantly inhibits the exocytosis of NPs from the cell. 

### 4.3. Enhancing the Radiation Effect by Restricting Cell Cycle Progression

Some clinically relevant chemotherapeutics such as DTX and PTX (members of the taxane family) can arrest the tumour cells in the most radiosensitive phase of the cell cycle, the G2/M phase [139,159,160]. During this phase, microtubule assembly is inhibited by DTX and PTX, restricting the cell from undergoing mitosis. The taxane family has been extensively studied in the literature and clinically employed for breast and other forms of cancer [6]. By their conjugation with lipids, proteins, and other targeting molecules, their tumour-cell-specific uptake and retention have been greatly improved. 

Transferrin (Tf) receptors are upregulated on the surfaces of cancer cells, which makes them potential targets for targeting ligands. Cui et al. [140] developed Tf-conjugated poly(lactide-co-glycolide) (PLGA) NPs loaded with PTX to target MCF-7 and U87 cancer cells. Their results showed that Tf-modified NPs greatly improved the cytotoxic effect of PTX compared to unmodified NPs. In addition, the modified NPs also exhibited greater intracellular accumulation around the nucleus. Similarly, Jose et al. [141] developed Tf-conjugated PLGA NPs loaded with DTX trihydrate to target MCF-7 cells. Their conclusions also reported that conjugated NPs demonstrated significantly improved cellular uptake and cytotoxicity compared to non-conjugated NPs. Other notable targeting ligands employed in the literature include FA, antibodies, aptamers, and peptides, among others [156]. 

In addition to improving the cytotoxic effect of taxanes on the cell cycle, their specific disruption at the radiosensitive G2/M phase indicates potential for combinatory treatments such as PTX + high-Z NPs. The combination of the two modalities could improve cell arrest by inhibiting mitosis and trapping the cells in the most radiosensitive phase of the cell cycle, enabling more effective radiotherapy treatments. A recent publication by Bromma et al. [142] evaluated the combinatory effects of GNPs functionalized with PEG+RGD with DTX on a two-dimensional monolayer and three-dimensional spheroids of LNCaP prostate and cervical HeLa cells. In most pre-clinical studies on cancer cell lines, 2D cell layers are employed; however, their conditions do not accurately represent those found within a tumour. Cells dosed with DTX and treated with GNPs had increased GNP cellular uptake compared to those without DTX treatment (*p* < 0.001) (Figure 9a,b). Cells incubated with DTX and GNPs were subsequently irradiated at 2, 5, and 10 Gy doses from a 6MV LINAC. The combination of DTX+GNPs+RT increased the DSBs by 91.6–109.9% compared to DTX alone in 2D monolayer samples. In the 3D spheroids, GNP+DTX+RT decreased the survival of the cancer cells by 20.9–31.4% compared to the control (Figure 9c). Their results demonstrate the synergistic effect of DTX+GNPs+RT on spheroids that better mimic the TME. 

## 5. Future Directions of Targeted Nanomaterials in Cancer Therapy

Current advancements in the clinical cancer treatment workflows typically involve the combination of current therapeutics with other targeted drugs. Tranilast (N-(3,4-Dimethoxycinnamoyl)anthranilic acid) is an anti-fibrotic drug that has been used in combination with doxorubicin—a clinically relevant chemotherapeutic drug—to reduce the stiffness of the stromal matrix [129]. It has also shown potential to target CAFs by inhibiting cross-talk with non-small-cell lung cancer cells [161]. A current Phase I/II clinical trial (NCT05626829) is recruiting to evaluate the effectiveness of Tranilast as a radiosensitizer for nasopharyngeal carcinoma, with completion aimed at Fall 2024. As discussed previously, another promising radiosensitizer, KORTUC, is also recruiting for a Phase II clinical trial (NCT03946202) with completion in early 2026. The study aims to use a novel gel containing hydrogen peroxide to improve the radiosensitization of breast cancer to a typical 3-week treatment plan. Another clinical trial with KORTUC that is not currently recruiting aims to evaluate the safety and efficacy of KRC-01 (NCT05570422). The study aims to perform both a Phase I and II trial with 60 patients receiving external beam RT and Cisplatin. Another promising radiosensitizer, NBTXR3 (HfO_2_ NP), is currently under several clinical trials for use with radiotherapy to target various forms of cancer, such as head and neck (NCT01946867), soft tissue carcinoma (NCT01433068), and lung (NCT04505267), among many others. 

The effective translation of targeted therapeutics on the TME into the clinic requires extensive hurdles to be overcome to ensure the safety and efficacy of the drugs with current treatments. For example, the effective dose administrable to humans can greatly differ from that delivered to mice; therefore, a clear understanding of the translatable and clinically relevant drug concentrations is required. As discussed in Section 4, the first avenue should be to closely mimic the tumour and its microenvironment using 3D spheroids and organoids, as this cannot be effectively achieved with 2D cell monolayers [138,142,162]. These additional results, prior to mouse irradiations and clinical trials, will help to develop more effective treatment plans incorporating the effect of the NP distribution in solid tumours. While many preliminary studies regarding the improved targeting of the tumour and its microenvironment exist, limited clinical studies of their use on a patient cohort have been found. Furthering the application of 3D spheroids in the TME and radiotherapeutics, their incorporation onto microfluidic chips that assay the tumour microenvironment during angiogenesis offers a promising avenue [163]. Microfluidics is a miniaturization technology incorporating multiple functions to evaluate small biological systems. While 3D spheroids alone demonstrate a significant step towards mimicking the tumour response, their incorporation with microfluidic chips could offer further insight into the tumour microenvironment, its response to therapeutic agents, and tumourigenesis. 

As discussed by Varzandeh et al. [164], many pathways exist by which combinatory treatments can be applied for cancer therapy. Given the clinically relevant chemotherapeutic drugs docetaxel, paclitaxel, and cabazitaxel, which restrict cell cycle progression by microtubule stabilization to the radiosensitive G2/M phase, their combined use with other radiosensitizers for radiotherapy treatment offers a unique advantage to enhance the therapeutic effect of radiation greatly. It was previously shown that the combination of GNPs with DTX demonstrated significantly greater cell killing than either modality alone [142]. By selectively targeting the tumour and its microenvironment, a significant reduction in drug dose and treatment prescription may be achievable, greatly reducing normal tissue toxicity. Certain barriers must be overcome to translate this technique to the clinic. First, significantly more pre-clinical studies should be performed to evaluate the combinatory effects of NPs with taxanes properly. The desired results will help to prescribe the effective dose and conjugation pairs for the effective targeting of different malignancies. Next, the proper scaling techniques to make the required amount of nanomaterials and drugs available, along with their synthesis, will have to be developed; these will depend on the associated clinically relevant dosage as found previously. 

Lastly, not explicitly discussed within this review is immunotherapy, an increasingly common cancer treatment aimed at improving the body’s natural immune response to the malignancy [165,166]. As already briefly described in this article, nanoparticles and other conjugate materials are already in pre-clinical workflows to reprogram recruited immune cells in the TME, demonstrating that nanomaterials can specifically increase the anti-tumour response by promoting lymphocyte recruitment. Research into their applications is growing exponentially and could reduce the complication rates associated with the toxicity inherent in chemotherapy and radiotherapy.

## 6. Conclusions

With the plateau of current chemotherapy and radiotherapy treatments stemming from normal tissue toxicity limitations, there is a requirement to develop adaptations and augmentations to the treatments to continue improving treatment plans and reported outcomes. In this review, we have discussed the ever-expanding research area of conjugated therapeutics with nanoparticles and other targeting ligands. Chemotherapeutics, immunotherapeutics, and radiotherapeutics offer significant combative advantages to patients suffering from malignant diseases. Using organic and inorganic compounds, these drugs can be targeted to specific binding domains within the tumour or the TME. By targeting the tumour microenvironment and other cellular pathways within the malignant growth, conjugated therapeutics are becoming increasingly effective additions to current cancer treatments and should continue to be studied.

## Figures and Tables

**Figure 1 pharmaceutics-16-00175-f001:**
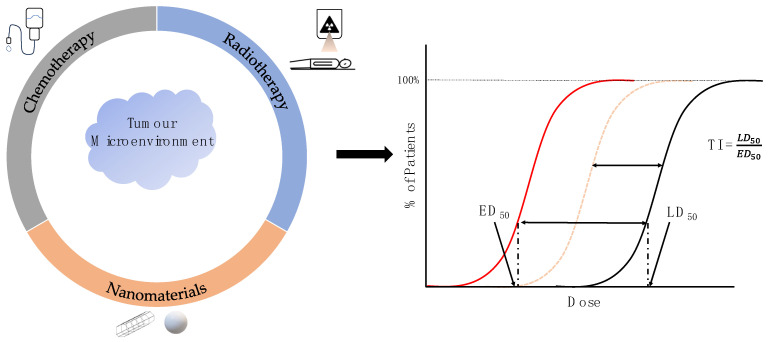
Cancer therapeutics targeting the tumour microenvironment. The TME is a pivotal contributor to the tumour’s ability to proliferate and metastasize. By employing nanomaterials and other targeted therapeutics towards the TME, the therapeutic index (TI) can be increased. The TI is defined as the ratio between the lethal dose to 50% of the population (LD_50_) and the effective dose to 50% of the population (ED_50_) of a specific treatment or drug. Described in this figure is the case where the ED_50_ decreases, as demonstrated by the dashed orange (without targeted therapeutics) and red lines (with target therapeutics); however, improving the TI can also be achieved by increasing the LD_50_. Note that the figure assumes that the addition of the therapeutic does not affect the LD_50_ similarly to the ED50; however, this may not be clinically feasible.

**Figure 2 pharmaceutics-16-00175-f002:**
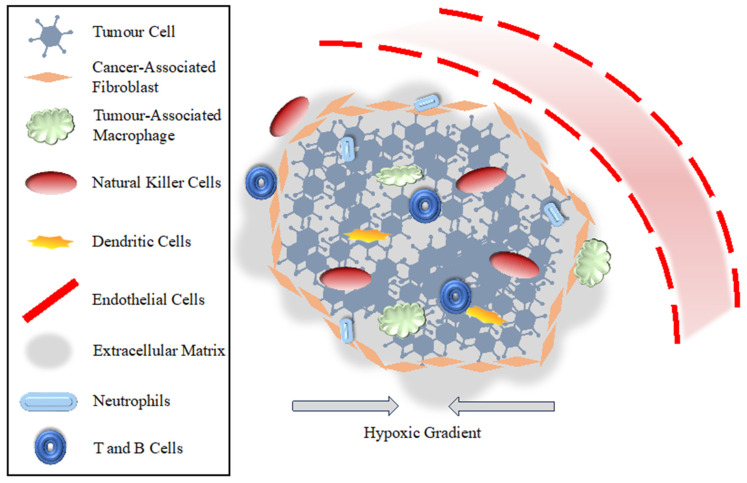
Pivotal contributors to the tumour microenvironment, its immune suppression and evasion, and progression and metastases. The hypoxic gradient stems from the inability of nutrients and oxygen to effectively penetrate the tumour growth, leading to a necrotic region within the centre, a quiescent region towards the edges, and a proliferation region at the edges.

**Figure 3 pharmaceutics-16-00175-f003:**
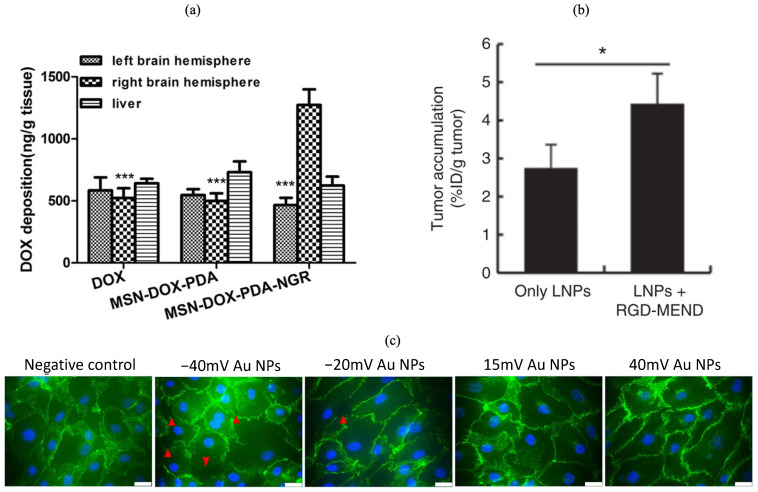
Target of endothelial cells via conjugated nanomaterials. Doxorubicin (DOX) deposition in tumour-bearing brain tissue and liver tissue with different nanoparticle formulations: mesoporous silica NPs (MSN), polydopamine (PDA) (**a**); accumulation of lipid NPs (LNPs) into RCC, OS-RC-2 bearing mice with or without RGD-MEND (**b**); immunofluorescent images of NP-induced endothelial leakage by GNPs with different surface charges, scale bar: 25 μm (**c**). (* indicates *p*-value < 0.05; *** indicates *p*-values < 0.001). (**a**) Adapted with permission [85]; (**b**) reprinted from Molecular Therapy, 24(12), Y. Sakurai, T. Hada, S. Yamamoto, A. Kato, W. Mizumura, and H. Harashima, “Remodeling of the Extracellular Matrix by Endothelial Cell-Targeting siRNA Improves the EPR-Based Delivery of 100 nm Particles”, 2090–2099, Copyright 2023, with permission from Elsevier. (**c**) Adapted with permission from [106]. Copyright 2023 American Chemical Society.

**Figure 4 pharmaceutics-16-00175-f004:**
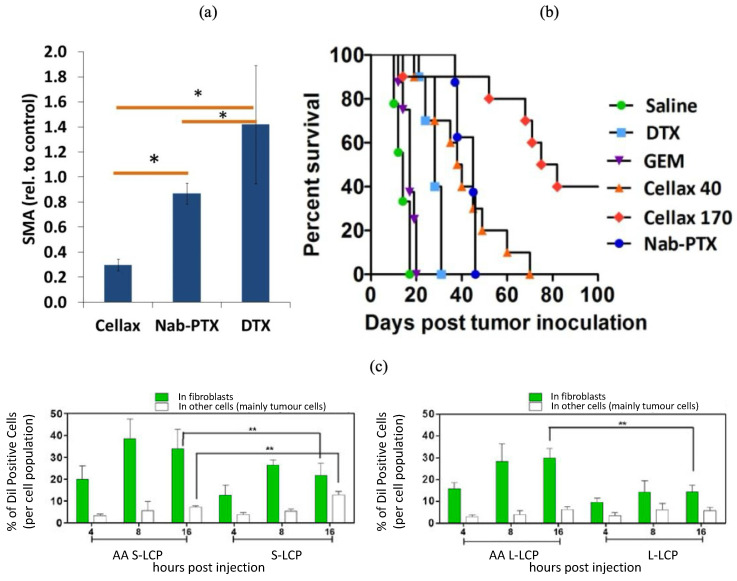
Cancer-associated fibroblasts and their targeting with nanomaterials. Smooth muscle actin (SMA) levels after tumours were treated with Cellax-DTX, Nab-PTX, and DTX (**a**); survival curves of mice treated with DTX, gemcitabine (GEM), Nab-PTX, or Cellax-DTX at concentrations of 40 or 170 mg DTX/kg (**b**); time-dependent cellular distribution of small (S) and large (L) LCPs within CAFs or other cells with or without anisamide (AA) (**c**). (* indicates *p*-value < 0.05; ** indicates *p*-value < 0.01). (**a**,**b**) Reprinted from [93], Copyright 2023, with permission from Elsevier. (**c**) Reprinted with permission from [91]. Copyright 2016 American Chemical Society.

**Figure 5 pharmaceutics-16-00175-f005:**
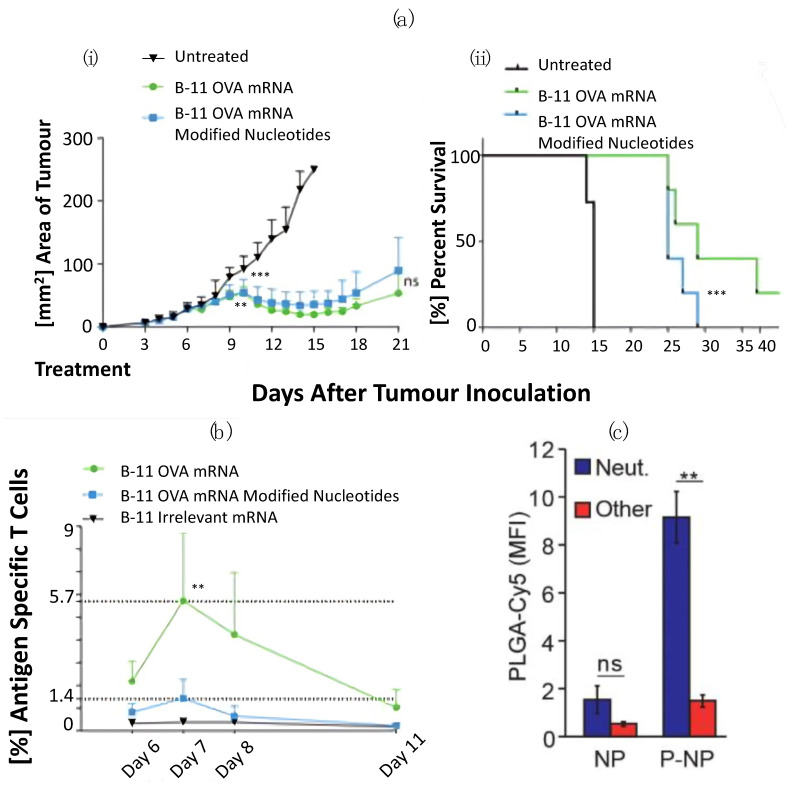
Reprogramming and targeting of the innate immune response towards malignancies via target nanomaterials. (**i**) LNP formulation B-11 induced potent in vivo anti-tumour immunity compared to untreated tumours; (**ii**) mice survival curves treated and untreated with LNP formulation B-11 from time of treatment to number of days after tumour inoculation (**a**); percentage of ovalbumin (OVA)-specific CD8 T cells after subcutaneous injection into mice (**b**); quantification via flow cytometry to detect PLGA-Cy5 of uncoated (NP) or LQI-coated NP (P-NP) binding to neutrophils (blue) or other white blood cells (red) (**c**). (“ns” indicates not significant; ** indicates *p*-value < 0.01; *** indicates *p*-value < 0.001). (**a**,**b**) Adapted with permission from [97]. Copyright 2023 American Chemical Society. (**c**) Copyright © 2022 [96].

**Figure 6 pharmaceutics-16-00175-f006:**
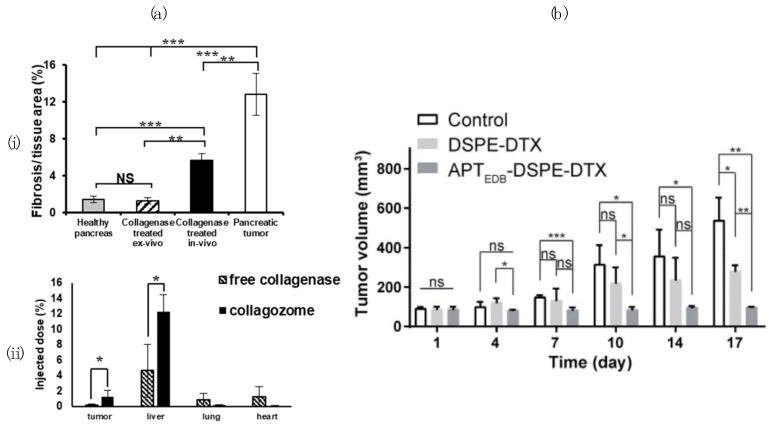
Targeting the extracellular matrix to improve nanomaterial uptake. Collagen density in a healthy pancreas, collagenase treated ex vivo and in vivo, and pancreatic tumour (i); biodistribution of GNPs to various organs following collagenase and collagozome treatment (ii) (* indicates *p*-value < 0.05; ** indicates *p*-value < 0.01; *** indicates *p*-values < 0.001) (**a**); anti-cancer effect on glioma tumour volume of NP formulation 1,2-dioctadecanoyl-sn-glycero-3-phosphoethanolamine (DSPE)-DTX and APT_EDB_-DSPE-DTX compared to the control group (**b**). (“ns” and “NS” indicate not significant; * indicates *p*-value < 0.05; ** indicates *p*-value < 0.01; *** indicates *p*-values < 0.001). (**a**,**b**) Reproduced with permission according to the Creative Commons Attribution license (https://creativecommons.org/licenses/by/4.0/ (accessed on 24 January 2024)).

**Figure 7 pharmaceutics-16-00175-f007:**
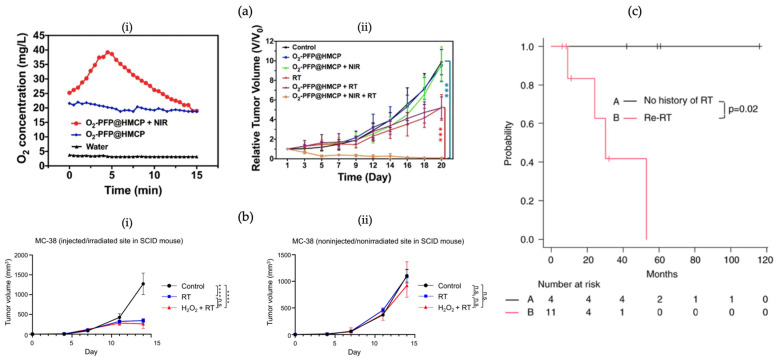
Stabilizing DNA damage and inducing hyperoxia via target nanoparticles. (**i**) O_2_ concentration within water with NP administration with or without laser irradiation; (**ii**) tumour growth curves of mice bearing U87MG tumours subjected to various combinations of NPs and irradiation (**a**); in vivo tumour volume of mouse models injected (**i**) or noninjected (**ii**) with H_2_O_2_ (KORTUC) and with RT compared to RT alone and control (**b**); comparison of local control rate with and without history of radiation of patients receiving interstitial brachytherapy with the administration of KORTUC (**c**). (“n.s.” indicates not significant; *** indicates *p*-values < 0.001; **** indicates *p*-value < 0.0001). (**a**) Adapted with permission from Biodegradable Hollow Mesoporous Organosilica Nanotheranostics for Mild Hyperthermia-Induced Bubble-Enhanced Oxygen-Sensitized Radiotherapy. Copyright 2018 American Chemical Society. (**b**) Adapted with permission according to the Creative Commons Attribution license (https://creativecommons.org/licenses/by-nc/4.0/ (accessed on 24 January 2024)). (**c**) Adapted with permission according to the Creative Commons Attribution license (https://creativecommons.org/licenses/by-nc-nd/4.0/ (accessed on 24 January 2024)).

**Figure 8 pharmaceutics-16-00175-f008:**
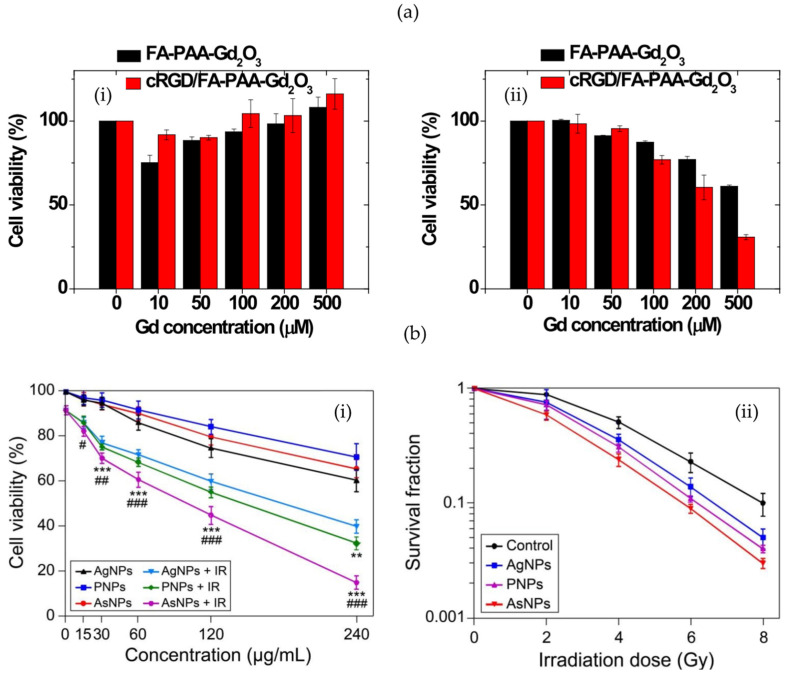
Metal nanoparticles improving efficacy of radiotherapy with targeted ligands. Cell viability of normal NCTC1469 cells (**i**) and U87MG tumour cells (**ii**) treated with FA-polyacrylic acid (PAA)-Gd_2_O_3_ and cRGD/ FA-PAA-Gd_2_O_3_ (**a**); (**i**) effects of AgNPs, PNPs, and AsNPs on C6 cell viability with or without irradiation; (**ii**) effects of AgNPs, PNPs, and AsNPs with irradiation on colony formation of C6 cells (**b**). (** indicates *p*-value < 0.01; *** indicates *p*-values < 0.001 when compared with the corresponding AgNPs treated group; # *p* < 0.05, ## *p* < 0.01, ### *p* < 0.001 when compared with the corresponding PNPs treated group). (**a**) Adapted with permission according to the Creative Commons Attribution license (https://creativecommons.org/licenses/by/4.0/ (accessed on 24 January 2024)). (**b**) Adapted with permission according to the Creative Commons Attribution license (http://creativecommons.org/licenses/by-nc/3.0/ (accessed on 24 January 2024)).

**Figure 9 pharmaceutics-16-00175-f009:**
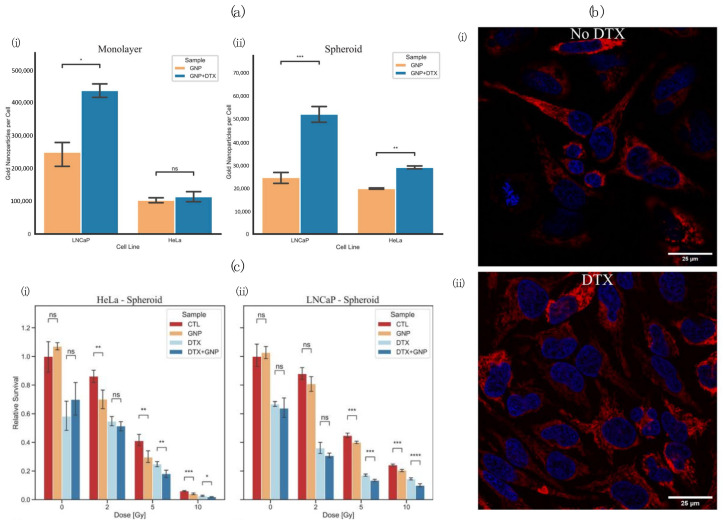
Combinatory effects of nanoparticles with taxane drugs and radiotherapy. GNP uptake within monolayer (**i**) and spheroid (**ii**) LNCaP and HeLa cells with and without DTX (**a**); confocal images of monolayer HeLa cells incubated with GNPs without (**i**) and with (**ii**) DTX (**b**); proliferation assay after 6 days following radiation for HeLa and LNCap cells irradiated from a 6 MV LINAC (**c**). (* indicates 0.01 < *p* < 0.05, ** indicates 0.001 < *p* < 0.01, *** indicates 0.0001 < *p* < 0.001, **** indicates *p* < 0.0001). (**a**–**c**) Adapted with permission according to the Creative Commons Attribution license (http://creativecommons.org/licenses/by/4.0/ (accessed on 24 January 2024)).

**Table 1 pharmaceutics-16-00175-t001:** A non-exhaustive list of targeting molecules conjugated to current therapeutics in cancer treatment discussed in this review article. Demonstrated in the table is the wide array of various receptors and targets, targeting molecules, and conjugated NPs in research workflows.

Tumour Microenvironment Component	Receptor/Target	Targeting Molecule	Conjugated NP	Reference
Tumour Endothelial Cells (TECs)	α(v)β(3) integrin	RGD	PEG-PLX-RGD	[15]
			RGD-MEND	[84]
	Aminopeptidase N (CD13)	NGR NGR-CPP	MSN-DOX-PDA-NGR	[85]
			pcCPP/NGR-LP	[86]
	Heparin sulfate	CGKRK	PC-NP-PTX	[87]
	Cyclooxygenase-2 enzyme		NS 398	[88]
	Enhanced permeability and retention effect		PLGA-ICG-PEI-Ang1@M	[89]
Cancer-Associated Fibroblasts (CAF)	Fibroblast activation protein (FAP)	mAB	PNP-D-mAB	[90]
	Sigma receptor	Anisamide	LCP NP	[91]
	PDGF-βR	pPB-HSA	PPB-HSA-IFN-γ	[92]
	α smooth muscle actin	PEG and acetylated carboxymethylcellulose	Cellax-DTX	[93]
Immune Cells	CX3CI1—TAMs		7C1-Axo-siCX3CL1	[94]
	CD206—TAMs	Man-PEG1k-DOPE (Mannose)	CaZol@ pMNPs	[95]
	CD177—neutrophils	LQI-tetramer	P-NP	[96]
	CD8+ T cells	mRNA vaccine	B-11	[97]
	TLR9—TAMs	CpG oligodeoxynucleotides	M2pep-rHF-CpG	[98]
Extracellular Matrix (ECM)	Collagen	Collogenase type-1	Collagozome	[99]
	Fibronectin	APT_EDB_	APTEDB-DSPE-DTX	[100]
			APT-NP-PTX	[14]

**Table 2 pharmaceutics-16-00175-t002:** A non-exhaustive list of radiopharmaceuticals and their mechanisms of action to improve local dose enhancement via separate pathways.

Nanoparticle	Mechanism of Action	Reference
DNA Damage Stabilizers and Hyperoxia-Inducing NPs		
Bi-SNPs	Limitation of HIF-α expression, induce release of NO	[131]
O_2_- PFP@HMCP TaOx@PFC-PEG	Improve oxygen concentration	[132] [133]
KORTUC	Maintain oxygen levels within malignant tissue containing larger amounts of hypoxic cells or anti-oxidative enzymes	[134]
High-Z Nanoparticles		
AGuIX	Increase photoelectric cross-section via Gadolinium	[135]
Polyacrylic acid-coated Gd_2_O_3_ + cRGD or FA	Improved toxicity and uptake with U87MG cells via dual conjugation	[136]
Ag NPs conjugated with PEG or As1411	Increase photoelectric cross-section via silver NPs	[137]
GNPs-PEG-RGD + DTX	Improved sensitization via cell cycle arrest and increased photoelectric cross-section from gold	[138]
GNP-PEG-RGD + DTX	Targeting of CAFs and improved NP retention	[139]
Cell Cycle Arrest		
Tf-PLGA + PTX Tf-PLGA + DTX	Tumour cell targeting via transferrin conjugation	[140] [141]
GNP-PEG-RGD + DTX	Improved sensitization via cell cycle arrest and increased photoelectric cross-section from gold	[142]

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
