# Peer review of "Improving the Efficacy of Common Cancer Treatments via Targeted Therapeutics towards the Tumour and Its Microenvironment"

_pharmaceutics, 2024, doi:10.3390/pharmaceutics16020175_

Round 1

Reviewer 1 Report

Comments and Suggestions for Authors

Comments:

1.     The quality of Figure 1 should be improved. The chemical formula is very vague. The English fonts in the picture are inconsistent. What are the symbols in the circle? For example, short lines, blue prisms, …etc.

2.     In Figure 2, what is the symbol representing Natural Killer Cells? They should be added to the picture.

3.     There are fewer references about recent three years cited in this review. The author should supplement it.

4.     This manuscript mainly summarized the strategies via targeted therapeutics towards the tumor and its microenvironment for improving the efficacy of chemotherapy and radiotherapy. However, immunotherapy was rarely mentioned. The author should supplement this aspect. Otherwise, the content of manuscript is insufficient to support the title of this review. Or, the author should change the title to make it more suitable for the content.

Comments on the Quality of English Language

The author should further refine the language.

Reviewer 2 Report

Comments and Suggestions for Authors

The manuscript titled “Improving the efficacy of cancer treatments via targeted therapeutics towards the tumour and its microenvironment” by Cecchi, D.; et al. is a Review work where the authors cover the state-of-the art and most recent advances in the design of conjugated nanomaterials with customized chemical drugs targeted depending the tumoral target. Many factors are address in this work like the tumour microenvironment conditions, the chemotherapeutic nature of the assessed agents and the biomolecular drivers of this disease.

However, it exists some points that need to be addressed (please, see them below detailed point-by-point) to improve the scientifc quality of the submitted manuscript paper before this article will be consider for its publication in Pharmaceutics.

1) SIMPLY SUMMARY and ABSTRACT (OPTIONAL). Maybe it may be more opportune the merge of these two sections in only one highlighting the most relevant points covered in this Review work.

2) KEYWORDS (OPTIONAL). The authors should consider to add the term “targeted therapies” in the keyword list.

3) INTRODUCTION. “The pandemic of increasingly common cancer-associated diseases (…) these malignancies” (lines 46-47). Please, recently reported quantitative data should be furnished about the worldwide burdens of cancer malignancies [1]

[1] https://doi.org/10.3322/caac.21763

4) “While surgery (…) and RT (…) via surgery” (lines 51-53). Please, the full-name of the terms should be defined the first time that they appear in the main manuscript body text. Then, the abbreviation should be placed between brackets.

5) PIVOTAL CONTRIBUTORS OF THE TUMOR MICROENVIRONMENT FOR TUMOURIGENESIS. “Carcinogenesis is a complex process that involves the accumulation of genomic and molecular mutations in healthy cells” (lines 147-148). Even if I agree with this statement provided by the authors, it may be advisable to briefly discuss about the existing clinical tools to diagnose and monitor the cellular genomic mutations. In this context, fluorescent in situ hybridization (FISH), optical genome mapping (OGM), and genome-wide association studies (GWAS) should be taken into account.

6) TARGETING THE TUMOUR MICROENVIRONMENT WITH CONJUGATED CHEMOTHERAPEUTICS AND NANOMATERIALS. In order to strengthen the importance of conjugated targeted nanoparticles for cancer therapies, it would be benefitial to compare them to other nanomaterials with magnetic [2] or electric [3] properties and remark their limitations (magnetic nanoparticle aggregation, citotoxicity, difficulty to control the magnetic and electrical performance according the nannoparticle size and dimensions, …).

[2] https://doi.org/10.3390/nano13182585

[3] https://doi.org/10.1016/j.jcis.2018.12.014

7) Figure 3, panel c (line 273). What is the lateral scale bar of the immunofluorescent images? This information should be stated at least in the respective figure caption. Same comment for the Figure 6, panel a (line 438, SEM images).

8) RADIOPHARMACEUTICALS AS LOCAL DOSE-ENHANCEMENT AGENTS. This section perfectly states the most recent advances in this field. No actions are requested from the authors.

9) FUTURE DIRECTIONS OF TARGETED NANOMATERIALS IN CANCER THERAPY. “As discussed (…) mimic the tumour and its microenvironment using 3-D spheroids and organoids, which cannot be effectively done with 2-D cell monolayers” (lines 672-674). I agree with the authors although the best approach is to combine the use of 3-D spheroids on microfluidic chips to assay the microenvironmental conditions that takes places during the angiogenesis. A brief statement should be added in this regard.

10) CONCLUSIONS. This section clearly remarks the most significant outcomes outlined in this Review work. Finally, the authors should carefully check the reference style according to the journal guidelines.

Comments on the Quality of English Language

The manuscript is generally well-written albeit it may be opportune if the authors take a final check to polish minor details susceptible to be improved.

Round 2

Reviewer 1 Report

Comments and Suggestions for Authors

The author should adjust the font of “Tumour Microenvironment” in Figure 1 to make it more aesthetically pleasing. 

Comments on the Quality of English Language

The quality of English fonts in Figures needs to be further improved.

Reviewer 2 Report

Comments and Suggestions for Authors

The authors covered all the suggestions raised by the Reviewers. For this reason, the scientific quality of this manuscript was greatly increased. Based on the significance of this work and the topic of Pharmaceutics journal, I warmly endorse this paper for further publication.

Author Response

No response to their comments is required.